# Longitudinal evidence over 2 years of the pandemic shows that poor mental health in people living with obesity may be underestimated

**Matthew J. Vowels**[1], **Laura M. Vowels**[2], **Jilly Gibson-Miller**[3]*

**1** Cognitive and Affective Regulation Laboratory (CARLA), Department of Social and Political Sciences, Institute of Psychology, University of Lausanne, Lausanne, Switzerland, **2** FAmily and DevelOpment Research Centre (FADO), Department of Social and Political Sciences, Institute of Psychology, University of Lausanne, Lausanne, Switzerland, **3** School of Education, The University of Sheffield, Sheffield, United Kingdom

* jilly.gibson@sheffield.ac.uk

**Data Availability Statement:** The data underlying the results presented in the study are available from https://osf.io/v2zur/files/osfstorage.

## Abstract

It is well-documented that people living with obesity are at greater risk of poorer mental health outcomes. The aim of our study was twofold: First, to examine the longitudinal trajectories of depression and anxiety in people living with obesity over two years across eight waves of a UK national COVID-19 survey (March 2020-March 2022) using smoothing-splines mixed-effects models. Second, to investigate participation effects via a missingness analysis to check whether survey attrition over time was related to participant characteristics. Trajectory models showed that those living with overweight and obesity consistently reported significantly higher rates of anxiety and depression compared to those in normal weight categories over two years. Our missingness analysis revealed that depression and anxiety predicted the likelihood of responding to the survey over time, whereby those reporting higher rates of depression and anxiety were less likely to respond to the survey. Our findings add to the literature surrounding the (long-term) link between living with obesity and poor mental health. Notably, our results suggest that people who have poorer mental health were less likely to participate in the survey. Thus, we conclude that it is likely that longitudinal population survey studies potentially underreport mental health problems over time and therefore the realistic impact of obesity on mental health outcomes may be underestimated.

## Introduction

The unprecedented circumstances surrounding the COVID-19 pandemic have disproportionately affected populations at high risk of serious outcomes, including those living with obesity (BMI ≥ 30 kg/m^2) who had higher odds of intensive care unit admissions, being placed on a mechanical ventilator, and death [1, 2]. As a consequence, individuals who were identified as obese in the United Kingdom were asked to 'shield' (i.e., completely avoid contact with other people) for 1.5 years between March 2020 and September 2021. Although there were variations in population mental health, research has shown that mental health problems in particular

**Funding:** Gibson-Miller, J. Mental health and well being in people living with overweight and obesity during the COVID-19 pandemic. Research England COVID 19 Recovery Fund. Jan-July 2022. £29,148.

**Competing interests:** The authors have declared that no competing interests exist.

groups of people, such as those with chronic health conditions, increased significantly during the pandemic [3]. In particular, self-isolation or shielding was especially detrimental to people's well-being [4, 5]. Pre-pandemic studies indicate a higher prevalence of mental health issues among individuals with obesity, whereby there is a bidirectional relationship between obesity and poor mental health (i.e., the likelihood of mental health problems increases for those with higher body weight) [6, 7]. For example, a study of 363 037 patients showed that higher body mass index (BMI) was associated with a higher likelihood of depression [7]. Likewise, a meta-review [8] found a statistically significant association between obesity and depression with those who had a BMI above 40, associated with greater odds of becoming depressed. One recent study has estimated the prevalence of depressive and anxiety disorders in obese patients at 29.23% and 25.56%, respectively [9].

The pandemic, intensifying this pre-existing vulnerability, has likely disproportionately amplified mental health problems for these individuals due to imposed self-isolation measures and general health risks associated with obesity [3]. Recent work [1] clearly illustrates that obesity is a risk factor for poorer mental health outcomes during the pandemic, including higher levels of depression and lower well-being, and more anxiety, fear, and worry about the pandemic, compared to pre-pandemic experiences [10–12].

In the present study, we examined the mental health (depression and anxiety) trajectories of people living with obesity compared to those who identified as overweight or normal weight across two years (eight points of data collection) of the COVID-19 pandemic (March 2020 – March 2022). We used survey data from a large nationally representative sample collected in the United Kingdom. Based on previous research, we expected that individuals living with obesity would report worse mental health outcomes during the pandemic followed by participants who reported being overweight with participants in the normal weight category reporting the lowest levels of depression and anxiety.

In addition to examining the mental health trajectories of people in different weight categories over the two-year period, we also aimed to understand how sample recruitment and the associated drop-out/top-up may affect the results within the study as well as potentially within other longitudinal survey studies examining obesity and mental health. When collecting longitudinal data, one must contend with participant drop-out/non-response, and/or top-up as a recruitment measure which helps compensate for drop-out by introducing additional participants to the study. Unfortunately, there is a risk that drop-out and top-up are non-random; sometimes drop-out occurs at times of difficulty for the participant, and these difficulties may be associated with the measures of interest for the study (in our case, the reasons for drop-out may be related to the participants' levels of depression or anxiety, or obesity itself). Similarly, top-up can also result in skew, because even if the original participant invitations are representative, the participants who finally agree to participate may not be. Furthermore, even if the new participants are representative, their inclusion may not complement the skew of the participants who dropped-out and may even exacerbate the imbalance that resulted from the non-random drop-out. As such, and to investigate whether drop-out or top-up could have resulted in a skew in the average levels of depression and anxiety at each timepoint, we attempted to predict how each participant who responded at least once throughout the study was likely to have responded at each of the timepoints separately.

## Materials and methods

### Participants and procedure

We conducted a secondary analysis of the longitudinal, COVID-19 Psychological Research Consortium Study (C19PRCS) data. A detailed methodological account is available elsewhere [13–15] but below we provide a brief description of the study methodology.

The data were collected through an internet-based survey fielded by UK survey company, Qualtrics. The survey included measures of socio-demographic characteristics, health characteristics and behaviour, knowledge, attitudes and beliefs in relation to COVID-19, mental health indicators, social attitudes, and psychological variables. Quota sampling was used to recruit a panel of adults who were nationally representative of the UK population in terms of age, sex, and household income. Participants were aged 18 years or older at the time of the survey, must have been able to complete the survey in English, and be resident in the UK. Adults provided informed consent before completing the survey online and were reimbursed by Qualtrics for their time. Ethical approval for this research was provided by a UK University Psychology department (Reference number: 033759). In the present study, we used data from Waves 1–8 spanning the first two years of the COVID-19 pandemic (March 2020 –March 2022) from all participants who self-reported being normal weight, overweight, or obese (underweight participants were not included in the study) in at least one of the waves in which measures of weight category were available (Waves 3, 5, and 8). The summary characteristics are reported in Table 1. On average, 8.6% of our sample reported being obese, 44.6% overweight and 46.8% normal weight (for context, current UK data [16] indicate that population estimates are 35.9% Obese and 37.9% overweight).

## Measures

Participants were asked to self-identify into one of four weight categories: underweight (4, removed from the analyses given the focus on obesity), normal (3), overweight (2), or obese

**Table 1. Participant summary characteristics at each timepoint.**

| Months | Group | Women | Men | Age | | | Dep. | GAD |
|---|---|---|---|---|---|---|---|---|
| | | | | Mean (SD) | Min | Max | M(SD) | M(SD) |
| 0 | Obese | 63 | 51 | 47.40(14.13) | 22 | 77 | 6.32(6.24) | 6.18(5.92) |
| 0 | Overweight | 325 | 322 | 51.69(14.68) | 20 | 83 | 4.88(5.67) | 4.77(5.33) |
| 0 | Normal weight | 341 | 377 | 46.30(15.14) | 18 | 83 | 4.23(5.56) | 4.25(5.37) |
| 1 | Obese | 44 | 45 | 50.29(13.90) | 23 | 77 | 7.40(6.56) | 5.76(5.71) |
| 1 | Overweight | 257 | 271 | 53.57(13.97) | 20 | 83 | 4.83(5.30) | 3.95(4.86) |
| 1 | Normal weight | 252 | 299 | 48.00(14.60) | 18 | 83 | 4.20(5.44) | 3.66(5.10) |
| 4 | Obese | 88 | 68 | 45.29(14.78) | 18 | 90 | 8.40(7.12) | 7.03(6.48) |
| 4 | Overweight | 392 | 398 | 49.06(15.68) | 18 | 83 | 6.48(6.62) | 5.09(5.70) |
| 4 | Normal weight | 505 | 493 | 42.64(15.44) | 18 | 89 | 5.39(6.08) | 4.34(5.22) |
| 8 | Obese | 124 | 85 | 48.94(13.66) | 20 | 90 | 8.81(7.39) | 6.97(6.49) |
| 8 | Overweight | 529 | 526 | 52.98(14.55) | 18 | 89 | 6.03(6.64) | 4.71(5.68) |
| 8 | Normal weight | 643 | 667 | 48.16(15.70) | 18 | 92 | 4.77(6.02) | 3.72(5.10) |
| 12 | Obese | 111 | 79 | 49.34(13.66) | 22 | 90 | 8.26(7.35) | 6.78(6.54) |
| 12 | Overweight | 513 | 512 | 53.25(14.48) | 20 | 89 | 5.79(6.25) | 4.56(5.52) |
| 12 | Normal weight | 586 | 644 | 49.14(15.45) | 18 | 92 | 4.58(5.78) | 3.65(5.10) |
| 17 | Obese | 69 | 52 | 45.66(14.06) | 18 | 77 | 8.18(6.63) | 6.32(6.04) |
| 17 | Overweight | 355 | 352 | 50.62(14.83) | 20 | 83 | 5.52(6.14) | 4.35(5.35) |
| 17 | Normal weight | 441 | 411 | 45.10(14.88) | 18 | 89 | 4.24(5.73) | 3.50(5.04) |
| 20 | Obese | 50 | 39 | 46.53(13.39) | 23 | 74 | 7.26(6.58) | 5.53(5.96) |
| 20 | Overweight | 265 | 285 | 51.08(14.89) | 20 | 82 | 5.58(6.45) | 4.46(5.65) |
| 20 | Normal weight | 326 | 327 | 46.19(14.78) | 18 | 89 | 4.00(5.43) | 3.43(4.85) |
| 24 | Obese | 90 | 79 | 42.17(14.49) | 18 | 75 | 11.13(7.21) | 8.35(6.22) |
| 24 | Overweight | 448 | 390 | 47.53(15.46) | 18 | 83 | 7.03(6.50) | 5.35(5.66) |
| 24 | Normal weight | 543 | 522 | 42.32(15.43) | 18 | 89 | 5.77(6.31) | 4.66(5.45) |

(1). Depressive symptoms were measured using the Patient Health Questionnaire PHQ-9 [17], a 9-item self-report measure that asks participants the degree to which they have been bothered by depressive symptoms in the last two weeks (ranging from 0 [not bothered at all] to 2 [bothered a lot]). Higher scores indicative of higher levels of depression and scores of between 8–11 indicate diagnostic levels of depression that may require psychological intervention [18].

Anxiety was measured using the Generalized Anxiety Disorder 7-item Scale (GAD-7; [19]). Participants indicated how often they had been bothered by each symptom over the past 2 weeks on a four-point Likert scale (0 = Not at all, to 3 = Nearly every day). Again, higher scores are indicative of higher levels of anxiety. The means and standard deviations for both scales for each weight category can be found in Table 1.

## Data analysis

**Multilevel spline trajectories.** First, we modelled the longitudinal trajectories of depressive symptoms and anxiety separately, and for each weight category group (obese, overweight, and normal weight). To model the longitudinal trajectories, we used smoothing-splines mixed-effects models, as provided in the R package *sme* [20]. The non-linear and mixed-effects components of the modelling technique allowed us to account for the auto-correlation present in repeated measures data, and to model the non-linear nature of each participant's anxiety and depression trajectories. We were thus able to model the trajectory of each participant separately (at the participant level) and to then create a group level average trajectory with confidence intervals determined by the inter-individual variability. While the interpretation of such models can be quite involved (the smoothing splines entail a substantial parameterization), with these models we were nonetheless able to visually represent the average trajectories for each group and to understand whether these groups differed significantly from one another over time. The two hyperparameters for the model $\lambda_\mu$ and $\lambda_v$ control the degree of non-linearity / the flexibility of the splines at the average and the individual levels, respectively. The values are non-negative real values between 0 and positive infinity, and higher values impose more constraints on the flexibility of the splines. For our analysis $\lambda_\mu = \lambda_v = 1$.

**Response / participation analysis.** Second, we investigated the potential drop-out and top-up participation effects, both of which have the potential to result in skewed or biased results. Fig 1 shows whether or not a participant responded at a particular time point (black) or not (white). Furthermore, Table 2 provides some quantification of how many participants responded in at least X timepoints (left) and how many participants responded at each timepoint (right). To make the predictions, we aggregated all people who participated at least once in the longitudinal study ($N$ = 4,143). In an ideal world, this aggregate sample would be close in its characteristics to the sample characteristics of participants at any single timepoint, and one would not be able to discern whether any individual participant in this aggregate was more or less likely to participate at any particular timepoint. This is because in an ideal world, top-up and drop-out would occur at random, and there would be nothing to differentiate the participation of each participant according to their characteristics. For each participant in this aggregate sample, we assigned a binary label for whether that participant responded (assign a 1) at each of the eight timepoints or not (assign a 0). We then attempted to correctly classify each participant's response for each timepoint using their age, gender, weight category (obese, overweight, or underweight), and average depression and average anxiety levels (averaged over the available measurements for these variables).

We used machine learning techniques to investigate the degree to which participants' characteristics were related to their participation to explore whether the trajectories themselves might be determined by these effects (in our case, the reasons for drop-out or top-up

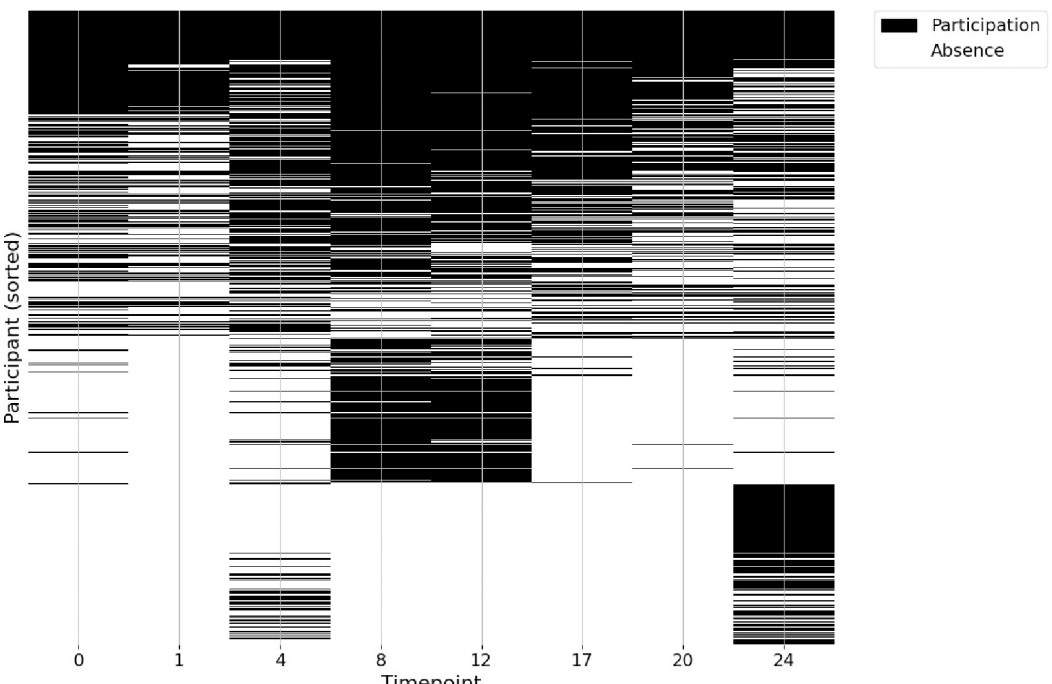

**Fig 1. Participant participation (black) or absence (white) sorted by degree of participation over time (high participation at the top, low participation at the bottom).**

participation may be related to the participants' levels of depression or anxiety). To investigate these effects, we used a gradient boosted decision tree [21]—specifically, one known as XGBoost classifier [22]—as implemented in the Scikit-Learn package in Python [23], to predict the missingness label for each participant in the aggregate sample, for each timepoint. We used a 66/33 train/test split to predict whether each participant who responded at least once throughout the study was likely to have responded at each of the timepoints separately. This train-test splitting helped us to cross-validate the trained model on unseen data. We then recorded the Balanced Accuracy performance of the XGBoost on these test data. Balanced Accuracy accounts for any imbalance in the dataset. Otherwise, predicting the majority category if the class imbalance is 90%/10% would result in an (unbalanced) accuracy of 90%, even though we would be misclassifying all the minority class examples. In contrast, in this example the Balanced accuracy would be 50%, which reflects the fact that we are not adequately discerning between the classes once we adjust for their relative prevalence in the data.

**Table 2. How many participants completed at least X number of timepoints (two leftmost columns), and how many participants there were at each timepoint (two rightmost columns).**

| No. of Timepoints | No. with at least X Timepoints of Participation | Month | No. of Participants (% of total participants) |
|---|---|---|---|
| 1 | 4143 | 0 | 1479 (36.7%) |
| 2 | 3097 | 1 | 1168 (28.2%) |
| 3 | 2150 | 4 | 1944 (46.9%) |
| 4 | 1756 | 8 | 2574 (62.1%) |
| 5 | 1452 | 12 | 2445 (59.0%) |
| 6 | 1082 | 17 | 1600 (38.6%) |
| 7 | 657 | 20 | 1292 (31.2%) |
| 8 | 317 | 24 | 2072 (50.0%) |

We used participants' age, gender, their weight category, and average depression and average anxiety levels (averaged over the available measurements for these variables) as predictors. Finally, to understand which of age, gender, weight category, depression, or GAD are useful in making these predictions (and thereby understand whether the drop-out or top-up is causing bias in the dataset), we used the "SHapley Additive exPlanations" package (SHAP) [24, 25]. The SHAP method derives from Lloyd Shapley's seminal work in the domain of game theory [26] and conceives of predictors as players in a collaborative game, where the goal is to maximise the predictive power of the algorithm. By exhaustively evaluating the impact that each individual predictor has in all possible combinations of predictors, the method can provide an estimation for the overall contribution of each predictor separately. It also provides us with a visualisation of how the distribution of predictor values pushes the model in one direction or the other. For example, high values of depression may push the model towards classification of the positive class, and vice versa for low values.

## Results

### Longitudinal trajectories of depression and anxiety

The longitudinal trajectories for depression and anxiety are shown in Fig 2. The results show that both anxiety and depression are significantly higher for people within the obese weight category (shown in red) than those in the overweight (shown in blue) or normal weight (green) categories. In general, people in the overweight category also had significantly higher levels of anxiety and depression than people of normal weight. For comparison, we have also provided the trajectories plotting the raw mean scores in S1 File (available on the OSF platform at https://osf.io/9t6ey/).

### Missingness analysis

The test-set balanced accuracy scores for the XGBoost algorithm classification of participant response at each timepoint are shown in Table 3, which shows that the algorithm can predict participation with a higher than chance accuracy–the balanced accuracy scores range from

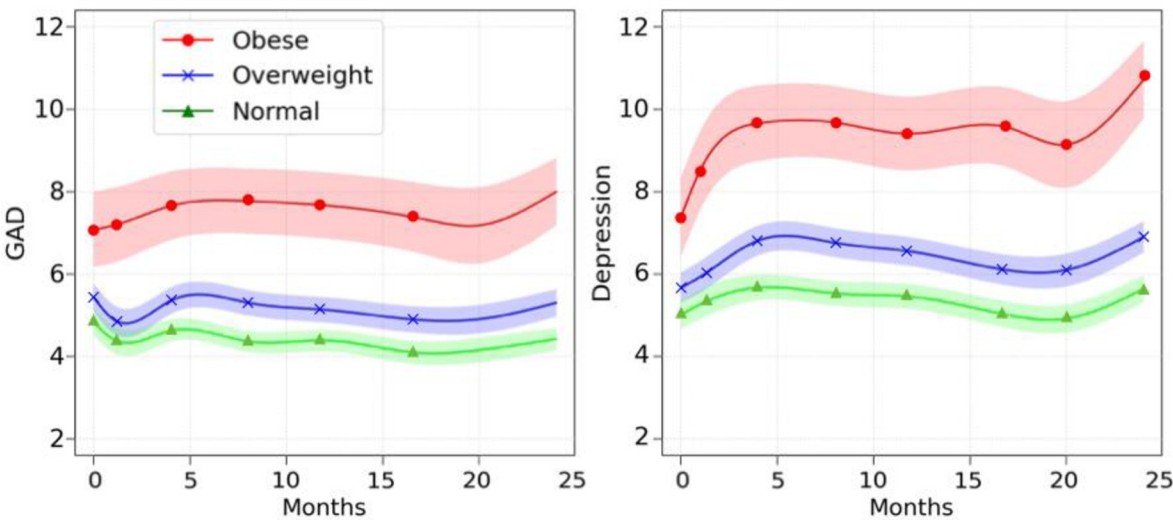

**Fig 2.** Smoothed-Spline Mixed-Model Average Trajectories with 95% Confidence Intervals for GAD (Left) and Depression (Right) for Each Weight Category.

**Table 3. Balanced accuracy scores for the XGBoost classification.**

| Month | Balanced Accuracy |
|---|---|
| 0 | 0.66 |
| 1 | 0.61 |
| 4 | 0.62 |
| 8 | 0.70 |
| 12 | 0.69 |
| 17 | 0.67 |
| 20 | 0.59 |
| 24 | 0.58 |

Balanced accuracy scores for the classification of each participant's participant at each of the 8 timepoints. Higher is better, and 0.5 represents chance level classification performance.

0.58 in the last timepoint, to 0.70 for month 8 (the balanced accuracy of random chance classification would be 0.50). Fig 3 shows the participants in each of the three weight categories (left hand side) and the relative impacts of each predictor (right hand side). The proportions are well balanced across the three weight categories, with no substantial qualitative differences apparent from this plot. For the predictors, depression was the most stably important predictor of participation across all timepoints, followed closely by anxiety. Age became substantially important as a predictor of response in timepoints 8 and 12 and was otherwise similar in its importance for predicting response to anxiety. Neither gender nor weight category were useful in predicting participant response, suggesting that these factors were well balanced across timepoints. The directions of these predictive effects can be observed in Fig 4. From these results we found that people higher in depression were consistently less likely to be classified as responding. This suggests that people high in depression either dropped-out or did not respond to invitations during top-up and/or subsequent recruitment periods.

## Discussion

The results of the present study showed that individuals living with obesity struggled more with poor mental health (depression and anxiety) compared to people who were overweight or normal weight over two years during the pandemic. Individuals who were overweight also reported higher scores on depression and anxiety compared to normal weight individuals, but

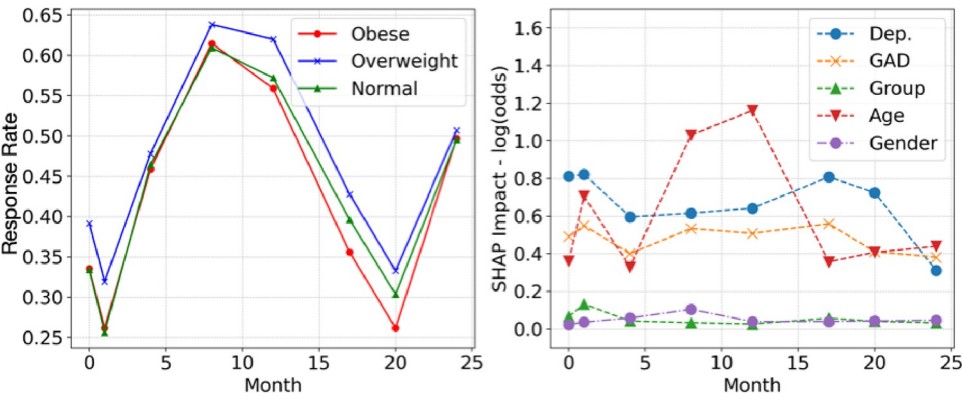

**Fig 3. Response rates and SHAP predictor importances.**

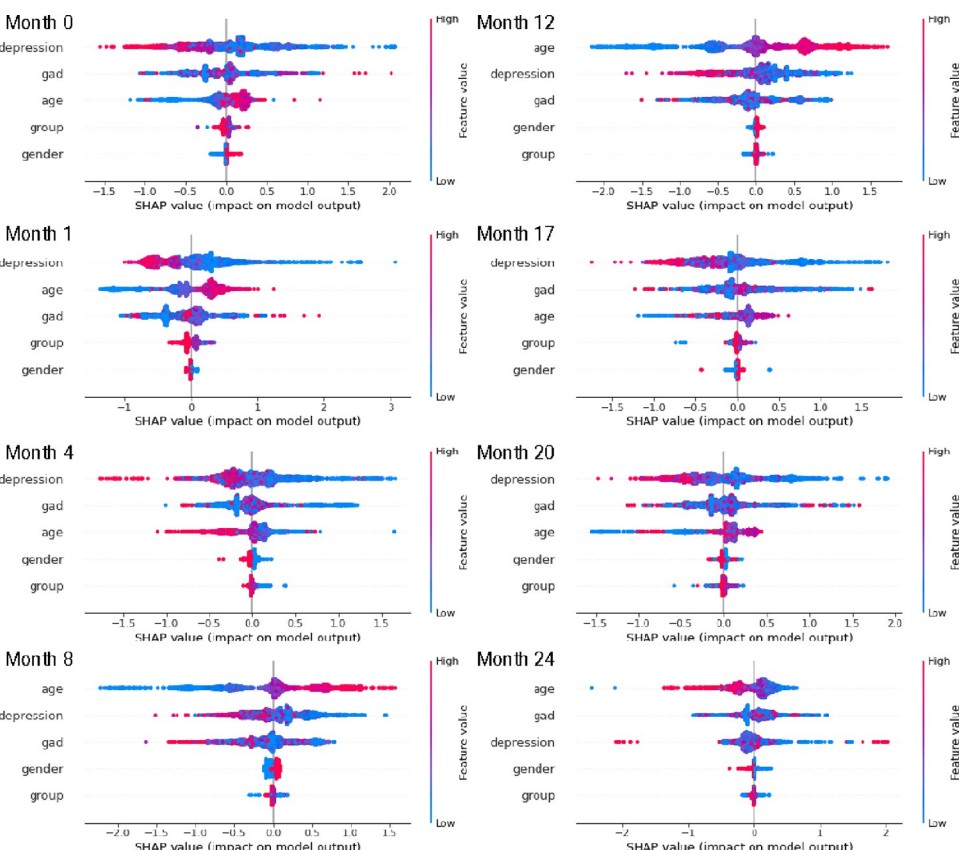

**Fig 4. SHAP per-predictor, per-datapoint model impact results.** For the classification of whether each participant in the aggregate sample responded at each of the 8 timepoints, we provide the SHAP impacts (in units of log(odds)) for each datapoint for each predictor, for each participant. For example, in Month 0, we see that high levels of depression (red), have a negative impact on the impact on the model output, which is measured in log(odds). In other words, a participant with high levels of depression is likely to be classified as not responding at this timepoint.

the difference was smaller. It is well-documented in the literature that people living with obesity are at greater risk of poorer mental health outcomes [e.g., 8] and we robustly illustrate that the pattern of comparison with those of lower weights is consistent over the long term. The scores on the mental health measures for those living with obesity reached levels that would be clinically indicative of mild-moderate anxiety [27] and moderate depressive disorder [18]. Within these data, we further illustrate the bi-directional relationship between obesity and poor mental health illustrated by Lavallee et al. and Moussa et al. [6, 7], whereby individuals in our sample who reported being heavier, showed worse mental health outcomes. This is one of the first studies of its kind to document in a large representative sample the association between obesity and mental health outcomes during the pandemic and the first to document the trajectories between weight categories over such a long time period.

The scale of the C-19 PRC study allowed us to investigate sample attrition, which is an important consideration for estimating population health from research. The results from the missingness analyses suggest that while there was no apparent bias in the weight categories (i.e., proportions in the weight categories remained fairly stable over time), people who were more depressed, and to a lesser extent more anxious, were less likely to participate in follow-up survey points. Therefore, the C-19 PRC survey, and others like it, are likely to actually

underreport mental health problems over time due to differential drop-out. This has important implications for representing the realistic impact of (for example) obesity on mental health outcomes, as distress is likely to be underestimated.

There are clear implications from our findings for the direction of future research efforts in this field. We recommend a focus on providing combined interventions for people living with obesity that address both health *and* mental health issues to interrupt the enduring bi-directional cycle. Indeed, it is now generally acknowledged in the field that conceptualising obesity as a chronic, relapsing and multifaceted health condition is useful in addressing the myriad of bio-psycho-social influences that contribute to the development and maintenance of obesity and reducing stigma [28]. Mohseni et al. [29] reported improved health, behavioural and psychological outcomes (including anxiety, depression, stress and disordered eating) from a combined lifestyle/CBT intervention, which occurred independently of weight loss. A recent review also identified the positive impact of behavioural weight management interventions on depression and mental-health related quality of life in adults [30]. However, there appears to be a general lack of evidence for interventions that encompass the multifactorial nature of obesity, with treatment success often only measured by weight loss [29].

Since our analysis shows that longitudinal population surveys may be underreporting mental health problems over time, and the rates reported in our sample indicated clinical levels of distress, there is an even more pressing need to understand mental health outcomes for those living with obesity. Reporting and mitigating the occurrence of missing data in survey research is a general scientific issue [see, for example, 31]. However, to improve the methodological robustness of longitudinal surveys, it may be important to explore factors that might assist or motivate participation over time in those struggling with mental health issues to increase ecological validity and provide a more representative picture. Researchers conducting longitudinal survey studies might consider running a missingness analysis as a matter of course, to check whether a particular subsection of the sample is dropping out, and then employ strategies that enable continued representation from that group.

There are some limitations that need to be considered in interpreting our findings. First, the weight category of participants was not measured at all timepoints and thus we may have missed some participants as they dropped in and out of the study over time. Additionally, it was not possible to track changes in self-categorisation over time. There is some evidence that rates of obesity increased over the pandemic [see, for example 16, 32] but this survey was limited in the time points the measure was taken (Waves 3, 5, and 8). Finally, the weight category was self-reported instead of being calculated based on participants' weight and height and therefore our categories could be inaccurate. Research shows that there are predictable inaccuracies in estimations of weight, such that those who are overweight tend to underestimate their weight status, as heavier weights become more 'normalised' [33]. Indeed, our data support findings that a substantial proportion of individuals with overweight or obesity may not identify accurately their weight status [34]. The average rates in each of the categories we report here indicate that significantly fewer people in our sample, compared to the general population, identified as being in the 'obese' category (8.6% in the sample, compared to 25.9% in the population) whereas, more people in our sample identified as being 'overweight' compared to population estimates (44.6% and 37.9% respectively). It is likely that, rather than individuals in our sample being on average of a lower weight than the general population, those of heavier weights have underestimated their weight and mis-categorised themselves, possibly because weight status is judged relative to visual body size norms (i.e., heavier body weights have become more normal and this has caused a recalibration of what is perceived as being 'normal' and 'overweight'; [34]).

## Conclusion

In conclusion, it is important that we remain vigilant to mental health problems experienced by people living with obesity and provide combined intervention where depression and anxiety are identified, as they are likely to be enduring. Further, since those with mental health problems are more likely to drop out of longitudinal survey studies, it is important to explore factors that might assist or motivate participation over time in under-reached groups and employ qualitative methodology to explore lived experience to gain a more in-depth picture of living with poor mental health and obesity.

## Author Contributions

**Conceptualization:** Matthew J. Vowels, Laura M. Vowels, Jilly Gibson-Miller.

**Data curation:** Matthew J. Vowels, Laura M. Vowels, Jilly Gibson-Miller.

**Formal analysis:** Matthew J. Vowels, Laura M. Vowels.

**Funding acquisition:** Jilly Gibson-Miller.

**Project administration:** Jilly Gibson-Miller.

**Writing – original draft:** Matthew J. Vowels, Jilly Gibson-Miller.

**Writing – review & editing:** Matthew J. Vowels, Laura M. Vowels, Jilly Gibson-Miller.

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
