## [Decision Letter · Decision Letter 0]

16 Jan 2024

PONE-D-23-34311Longitudinal evidence over 2 years of the pandemic shows that poor mental health in people living with obesity may be underestimatedPLOS ONE

Dear Dr. Gibson-Miller,

Thank you for submitting your manuscript to PLOS ONE. After careful consideration, we feel that it has merit but does not fully meet PLOS ONE’s publication criteria as it currently stands. Therefore, we invite you to submit a revised version of the manuscript that addresses the points raised during the review process.

I particularly agree with the comments about the need to contextualize the theory underpinning the research question a little better. And above all, the need to expand the information on the C19PRC study. This I am convinced that will reinforce the strength of your contribution.

We look forward to receiving your revised manuscript.

Kind regards,

Jordi Gumà, Ph.D.

Academic Editor

PLOS ONE

Reviewers' comments:

Reviewer's Responses to Questions

**Comments to the Author**

1. Is the manuscript technically sound, and do the data support the conclusions?

Reviewer #1: Partly

Reviewer #2: Yes

Reviewer #3: Yes

2. Has the statistical analysis been performed appropriately and rigorously? 

Reviewer #1: I Don't Know

Reviewer #2: Yes

Reviewer #3: Yes

3. Have the authors made all data underlying the findings in their manuscript fully available?

Reviewer #1: Yes

Reviewer #2: Yes

Reviewer #3: Yes

4. Is the manuscript presented in an intelligible fashion and written in standard English?

Reviewer #1: Yes

Reviewer #2: Yes

Reviewer #3: Yes

5. Review Comments to the Author

Reviewer #1: I read this manuscript with manuscript ID: PONE-D-23-34311 for PLoS One with interest. The title is revealing and making a very strong claim (“Longitudinal evidence over 2 years of the pandemic shows that poor mental health in people living with obesity may be underestimated”). My main issue is that this manuscript was clearly intended as a “brief report” and not as a regular “research article”. As such, many important details are missing or banished to the supplementary materials (which I had no access to for some reason). My recommendation would be to expand the manuscript, specifically:

1. Expand the introduction with a literature review to strengthen your claim that the mental health burden is underestimated in the target population. If possible, estimates of the depressive and anxiety symptoms in the population of people with obesity should be stated explicitly (perhaps a range of estimates if no meta-analysis is available).

2. Please move Table 1 describing your sample from the supplements to the main body of the manuscript.

3. Please describe the C19PRC study. The references are great for further reading but some details of the sample are also necessary in the manuscript so your readers understand the sampling process, generalisability and validity of your results.

4. Please provide all information necessary to replicate your model. (Feel free to ignore this comment if your R code included with the supplementary materials.)

5. Could you please add drop-out rates per time point and to what extent did the study suffer from intermittent missingness?

6. Please discuss how your results fit in with or are different from other studies in the literature and also discuss how the chosen analyses may have impacted your findings.

Other comments (in no particular order):

7. It is unclear to me if the C19PRC study used a probabilistic sampling method and was representative of the population the sample was drawn from or not. Please state this explicitly in the methods. If the sample is non-probabilistic, then you should consider selection bias and how it could have affected the results.

8. Which countries participated in the study? How many participants were included from each country? Was the sample size (which could be stated in the main body of the text) considered in the analyses?

9. Do you have any means to verify the self-reported weight categories?

10. Do you have participants with lower-than-normal BMI? How many? How does merging this group with the “normal” BMI group affect your results?

11. Did you use the PHQ-9 and GAD-7 total scores as outcome for the mixed models? If so, were the residuals normally distributed?

12. Line 78: PHQ-9 measures depressive symptoms and not depression. I would recommend making this distinction clear in the manuscript.

13. Line 84: PHQ-9 and GAD-7 are both very skewed in the general population due to a flooring effect. Means and standard deviations are not very useful to describe the centre and spread of the distribution. I would recommend adding median, IQR and range.

14. To obtain the PHQ-9 and GAD-7 total scores, did you use only observations with all items, or did you use an imputation method if there was item-level incompleteness?

Finally, I would like to add that despite the numerous comments, I believe this paper could be an important contribution to the literature. I especially like the analysis of the missing values.

Reviewer #2: This manuscript describes the longitudinal trajectories of depression and anxiety during the COVID pandemic and their association with the weight status, by reporting 8-wave UK national data. The authors should be commended for their focus on the link between obesity and mental health during the pandemic. However, despite the good methodology and data analysis, the study has a few shortcomings that need to be addressed on which I elaborate below.

It is unclear how the current study addressed the causal link between obesity and mental health. In my view, the current results only show that individuals with obesity reported higher levels of depression and anxiety than overweight or normal weight people. A cross-lagged panel analysis could be more suitable if the goal of the study is to examine whether the obesity status predicts depression and anxiety or vice versa. Please report the aim of this study with greater clarity.

Related to the above, what is the rationale for investigating drop-out and top-up participation effect? We have no information on that in the Introduction, and a clear aim for this point is lacking.

The description of the characteristics of the sample is poorly reported. Although the procedure of the study is reported in previous papers, it is important to describe how participants were selected and recruited online. Is it a nationally representative sample?

Moreover, we have no information on demographics such as nationality (or minoritized identity groups), income, health condition, access to care, as well as data on COVID-related restrictions (i.e. lockdown), number of infections/deaths. Given that these data varied across countries and regions during the pandemic, it is important to see if some of these variables had an impact on psychological distress.

It is also important to know how many individuals reported an obesity condition before the pandemic, given the meta-analytic evidence of an increase of weight problems among the general population during the COVID outbreak.

Moreover, we know that psychological distress among the obese population can be linked to binge eating behaviors. Do the authors have data on eating symptoms or dysfunctional eating behaviors for this sample during the pandemic?

My main concern is that participants were asked to self-identify into weight categories, but we have no validity checks on these responses. For example, did participants rate their weight according to clear criteria? How can an individual distinguish between the categories of being overweight and being obese? Participants did not report their weight and height, and the weight status seems unclear.

The results section is well-reported.

The discussion section can be enriched in a number of ways. For example, what the current findings add to the literature on obesity and mental health during the pandemic? There are a number of studies and meta-analyses on this topic that could be used to improve this section.

It may be worthwhile to expand on how the current data on mental health trajectories add to the literature on mental health distress during the pandemic. This research topic has been addressed extensively over the last 2 years and should be discussed in more detail.

Reviewer #3: In the manuscript PONE-D-23-34311 the Authors examined the trajectories of depression and anxiety in people with obesity over two years across eight waves of a UK national COVID-19 survey (March 2020-March 2022). Trajectory models showed that those overweight and with obesity consistently reported significantly higher rates of anxiety and depression compared to those in normal weight categories over two years. The analysis revealed that depression and anxiety predicted the likelihood of responding to the survey over time, whereas those reporting higher rates of depression and anxiety were less likely to respond to the survey.

I believe that the study undertaken by the authors is valuable, relevant and topical to the subject of public health. Their findings may not only deepen the existing knowledge of the potential mental health problems of people living with obesity, but also contribute to the search for practical solutions to improve their quality of life. My general impression of the manuscript was positive. I recommend the manuscript for publication with a few modifications:

• I suggest structuring the Methods section with subheadings for easier navigation and greater clarity.

• Please complete the Methods section with information on the characteristics of the study group.

• Please include information on Ethical Approval.

• The discussion seems too general and needs refinement. First of all, I suggest creating connections to previous research and ongoing discourse.

6. PLOS authors have the option to publish the peer review history of their article (what does this mean?). If published, this will include your full peer review and any attached files.

Reviewer #1: No

Reviewer #2: No

Reviewer #3: No

---

## [Author Response · Author response to Decision Letter 0]

12 Mar 2024

Reviewer #1: I read this manuscript with manuscript ID: PONE-D-23-34311 for PLoS One with interest. The title is revealing and making a very strong claim (“Longitudinal evidence over 2 years of the pandemic shows that poor mental health in people living with obesity may be underestimated”). My main issue is that this manuscript was clearly intended as a “brief report” and not as a regular “research article”. As such, many important details are missing or banished to the supplementary materials (which I had no access to for some reason). My recommendation would be to expand the manuscript, specifically:

1. Expand the introduction with a literature review to strengthen your claim that the mental health burden is underestimated in the target population. If possible, estimates of the depressive and anxiety symptoms in the population of people with obesity should be stated explicitly (perhaps a range of estimates if no meta-analysis is available).

We have expanded the introduction and refined our research questions to make our claims clear. We believe that the findings of our study indicate that the mental health burden in this population may well be underestimated using longitudinal survey methodology. Whilst findings in the current literature indicate clearly that this burden is significant, and our work supports that, few studies have investigated attrition in this way. Our revised introduction explains this more clearly.

2. Please move Table 1 describing your sample from the supplements to the main body of the manuscript.

Thanks, this Table has now been moved to the main manuscript.

3. Please describe the C19PRC study. The references are great for further reading but some details of the sample are also necessary in the manuscript so your readers understand the sampling process, generalisability and validity of your results.

We have now added a paragraph in the Methods section that describes the C19PRC study in more detail. 

4. Please provide all information necessary to replicate your model. (Feel free to ignore this comment if your R code included with the supplementary materials.)

We have included the R and Python scripts in supplementary material.

5. Could you please add drop-out rates per time point and to what extent did the study suffer from intermittent missingness?

As the dynamics of participation can be difficult to summarize succinctly, we have provided this information in the manuscript (see Fig 1 and Table 2, below). As can be seen, the dropout/participation dynamics are complex. We hope that an evaluation of such dynamics can be encouraged in future work in the area.

 

Figure 1 

Participant participation (black) or absence (white) sorted by degree of participation over time (high participation at the top, low participation at the bottom).

Table 2 

How many participants completed at least X number of timepoints (two leftmost columns), and how many participants there were at each timepoint (two rightmost columns).

No. of Timepoints No. with at least X Timepoints of Participation Month No. of Participants

(% of total participants)

1 4143 0 1479 (36.7%)

2 3097 1 1168 (28.2%)

3 2150 4 1944 (46.9%)

4 1756 8 2574 (62.1%)

5 1452 12 2445 (59.0%)

6 1082 17 1600 (38.6%)

7 657 20 1292 (31.2%)

8 317 24 2072 (50.0%)

6. Please discuss how your results fit in with or are different from other studies in the literature and also discuss how the chosen analyses may have impacted your findings.

We have now expanded the discussion section to include more context to our findings.

Other comments (in no particular order):

7. It is unclear to me if the C19PRC study used a probabilistic sampling method and was representative of the population the sample was drawn from or not. Please state this explicitly in the methods. If the sample is non-probabilistic, then you should consider selection bias and how it could have affected the results.

The information we have added to the Methods section covers this point. We state that: “Quota sampling was used to recruit a panel of adults who were nationally representative of the UK population in terms of age, sex, and household income.”

Of course it is possible that a different sampling method may impact the results, because it may change the substrata of participants who do or do not respond at particular time points. However, to say exactly in which direction the results would change is difficult and would only be speculative, therefore we have not expanded on this in the manuscript.

8. Which countries participated in the study? How many participants were included from each country? Was the sample size (which could be stated in the main body of the text) considered in the analyses?

The data we analysed here was from the UK arm of the study, so includes participants from England, Scotland, Wales, and Northern Ireland. For the analyses, the full sample size for anyone who completed at least one time point was used (N=4143). The multilevel spline method takes advantage of data wherever it is available. For the missingness analysis, for each participant, a label for participant was assigned for each timepoint, thus even if a participant only participated once, we could still make predictions using the information collected at that time point to predict their participation at other timepoints.

The following sample size information has been added to the manuscript in the data analysis section:

“To make the predictions, we aggregated all people who participated at least once in the longitudinal study (N=4143).”

See also Table 2, which provides more information on the dropout/participation rates at each timepoint.

9. Do you have any means to verify the self-reported weight categories?

We have utilised self-reported data on weight category as we were unable to take verifiable measures of BMI due to the design of the study. Research shows that there are inaccuracies in self-perception of weight, such that it is likely that our sample have underestimated their body size. We discuss this literature and acknowledge this as a limitation of the study in the revised discussion section. 

10. Do you have participants with lower-than-normal BMI? How many? How does merging this group with the “normal” BMI group affect your results?

There are 160 people in the sample who identified as being underweight, but these participants were not included as part of the original analyses. If we combine the underweight with the normal weight, the results barely change. For example, here are the balanced accuracies for when the two lowest weight classes are combined:

Month Balanced_Acc

0 0.65

1 0.60

4 0.64

8 0.70

12 0.71

17 0.70

20 0.61

24 0.58

Compared with the original:

Month Balanced Accuracy

0 0.66

1 0.61

4 0.62

8 0.70

12 0.69

17 0.67

20 0.59

24 0.58

Similarly, the results for the important predictors in the combined sample:

are very similar to the original;

11. Did you use the PHQ-9 and GAD-7 total scores as outcome for the mixed models? If so, were the residuals normally distributed?

Yes, the summary scores were used for the mixed models. The residuals are not normally distributed - see example diagnostic plot for the depression outcome of the normal weight group:

In our experience, this is quite common with such models, particularly when paired with real-world data which, as the reviewer highlights below, are often skewed by floor or ceiling effects. 

However, our goal is not to derive and interpret parameter estimates for these models (doing so is already non-trivial given the non-linear nature of the models), but to fit a smooth trajectory to the depression and GAD scores over time. This is done to broadly characterise the differences between the groups, and we also provide the raw mean scores for comparison in the supplementary material. As such, given that we are not interpreting the coefficients of these models (e.g. with respect to significance) our view is that the non-normality of the errors does not impact the principal take-away regarding the average differences in perceived weight class between groups, over time.

12. Line 78: PHQ-9 measures depressive symptoms and not depression. I would recommend making this distinction clear in the manuscript.

Thanks, we have changed this accordingly throughout the manuscript.

13. Line 84: PHQ-9 and GAD-7 are both very skewed in the general population due to a flooring effect. Means and standard deviations are not very useful to describe the centre and spread of the distribution. I would recommend adding median, IQR and range.

Thanks for the suggestion, we have added two tables to the Supplementary material including the median, IQR, and range for each month, for each weight category, for depression and anxiety separately (Table S3 and S4, respectively).

14. To obtain the PHQ-9 and GAD-7 total scores, did you use only observations with all items, or did you use an imputation method if there was item-level incompleteness?

As this was a secondary data analysis, we used the publicly available data set, which included completed computation methods for missing data.

Finally, I would like to add that despite the numerous comments, I believe this paper could be an important contribution to the literature. I especially like the analysis of the missing values.

Many thanks for your positive feedback!

Reviewer #2: This manuscript describes the longitudinal trajectories of depression and anxiety during the COVID pandemic and their association with the weight status, by reporting 8-wave UK national data. The authors should be commended for their focus on the link between obesity and mental health during the pandemic. However, despite the good methodology and data analysis, the study has a few shortcomings that need to be addressed on which I elaborate below.

1. It is unclear how the current study addressed the causal link between obesity and mental health. In my view, the current results only show that individuals with obesity reported higher levels of depression and anxiety than overweight or normal weight people. A cross-lagged panel analysis could be more suitable if the goal of the study is to examine whether the obesity status predicts depression and anxiety or vice versa. Please report the aim of this study with greater clarity.

 We agree with the reviewer and have removed the comment about causality from the introduction. We believe the aims of the study are now clearer in the revised introduction. 

2. Related to the above, what is the rationale for investigating drop-out and top-up participation effect? We have no information on that in the Introduction, and a clear aim for this point is lacking.

Indeed, we have revised the introduction to cover this point in more detail.

3.The description of the characteristics of the sample is poorly reported. Although the procedure of the study is reported in previous papers, it is important to describe how participants were selected and recruited online. Is it a nationally representative sample?

Please also see response to reviewer 1, point 3. We have now included more information about the C-19 PRC study in the methodology section of the paper. Yes, the sample was selected to be representative of the UK population. We have also provided a Table in the manuscript (Table 1) describing the sample.

4. Moreover, we have no information on demographics such as nationality (or minoritized identity groups), income, health condition, access to care, as well as data on COVID-related restrictions (i.e. lockdown), number of infections/deaths. Given that these data varied across countries and regions during the pandemic, it is important to see if some of these variables had an impact on psychological distress.

First, to be clear, our study reports on UK data only and therefore no comparisons across countries are presented. As we took our sample across 8 waves of data and the sample would have varied over time, we felt that presenting this demographic data would be unnecessarily complex and not add to the study presented here. However, we have added summary data for anxiety and depression scores to Table 1 (please also see response to Reviewer 1, point 2).

5. It is also important to know how many individuals reported an obesity condition before the pandemic, given the meta-analytic evidence of an increase of weight problems among the general population during the COVID outbreak.

We have added population level estimates of overweight and obesity pre-and post-pandemic in our revised introduction and discussion sections. We discuss the rates of overweight and obesity in relation to our sample and acknowledge the limitations in the timing of our measurements.

6. Moreover, we know that psychological distress among the obese population can be linked to binge eating behaviors. Do the authors have data on eating symptoms or dysfunctional eating behaviors for this sample during the pandemic?

Unfortunately, we do not have data on eating behaviour from the C-19 PRC study.

7. My main concern is that participants were asked to self-identify into weight categories, but we have no validity checks on these responses. For example, did participants rate their weight according to clear criteria? How can an individual distinguish between the categories of being overweight and being obese? Participants did not report their weight and height, and the weight status seems unclear.

Please also see response to reviewer 1, point 9. The reviewer is correct, that we were unable to verify the weight categories chosen by participants due to the design of the study. The categorisation relied solely on self-perception. Research shows that there are inaccuracies in self-perception of weight, such that it is likely that our sample have underestimated their body size. We discuss this literature and acknowledge this as a limitation of the study in the revised discussion section. 

8. The results section is well-reported.

Thank you.

9. The discussion section can be enriched in a number of ways. For example, what the current findings add to the literature on obesity and mental health during the pandemic? There are a number of studies and meta-analyses on this topic that could be used to improve this section.

10. It may be worthwhile to expand on how the current data on mental health trajectories add to the literature on mental health distress during the pandemic. This research topic has been addressed extensively over the last 2 years and should be discussed in more detail.

The discussion section has now been expanded to incorporate these and the other reviewers' suggestions.

Reviewer #3: In the manuscript PONE-D-23-34311 the Authors examined the trajectories of depression and anxiety in people with obesity over two years across eight waves of a UK national COVID-19 survey (March 2020-March 2022). Trajectory models showed that those overweight and with obesity consistently reported significantly higher rates of anxiety and depression compared to those in normal weight categories over two years. The analysis revealed that depression and anxiety predicted the likelihood of responding to the survey over time, whereas those reporting higher rates of depression and anxiety were less likely to respond to the survey.

I believe that the study undertaken by the authors is valuable, relevant and topical to the subject of public health. Their findings may not only deepen the existing knowledge of the potential mental health problems of people living with obesity, but also contribute to the search for practical solutions to improve their quality of life. My general impression of the manuscript was positive. I recommend the manuscript for publication with a few modifications:

We thank the reviewer for their positive assessment of the paper and suggestions for revision.

1. I suggest structuring the Methods section with subheadings for easier navigation and greater clarity.

Thanks, we have changed this accordingly.

2. Please complete the Methods section with information on the characteristics of the study group.

Please see response to reviewer 2, point 4. 

3. Please include information on Ethical Approva

---

## [Decision Letter · Decision Letter 1]

4 Jun 2024

Longitudinal evidence over 2 years of the pandemic shows that poor mental health in people living with obesity may be underestimated

PONE-D-23-34311R1

Dear Dr. Gibson-Miller,

We’re pleased to inform you that your manuscript has been judged scientifically suitable for publication and will be formally accepted for publication once it meets all outstanding technical requirements.

Kind regards,

Jordi Gumà, Ph.D.

Academic Editor

PLOS ONE

Additional Editor Comments (optional):

Reviewers' comments:

Reviewer's Responses to Questions

**Comments to the Author**

1. If the authors have adequately addressed your comments raised in a previous round of review and you feel that this manuscript is now acceptable for publication, you may indicate that here to bypass the “Comments to the Author” section, enter your conflict of interest statement in the “Confidential to Editor” section, and submit your "Accept" recommendation.

Reviewer #3: All comments have been addressed

2. Is the manuscript technically sound, and do the data support the conclusions?

Reviewer #3: Yes

3. Has the statistical analysis been performed appropriately and rigorously? 

Reviewer #3: Yes

4. Have the authors made all data underlying the findings in their manuscript fully available?

Reviewer #3: Yes

5. Is the manuscript presented in an intelligible fashion and written in standard English?

Reviewer #3: Yes

6. Review Comments to the Author

Reviewer #3: I would like to thank the Authors for considering and responding to my comments. I have reviewed their responses and the revised manuscript, and I have no further concerns to raise. I think this is a good study with good statistical support. This current version of the manuscript meets the criteria of the PLOS ONE, and I would support its publication based on this revision.

7. PLOS authors have the option to publish the peer review history of their article (what does this mean?). If published, this will include your full peer review and any attached files.

Reviewer #3: No

---

## [Editor Report · Acceptance letter]

18 Jun 2024

PONE-D-23-34311R1 

PLOS ONE

Dear Dr. Gibson-Miller, 

I'm pleased to inform you that your manuscript has been deemed suitable for publication in PLOS ONE. Congratulations! Your manuscript is now being handed over to our production team.

Kind regards, 

on behalf of

Dr. Jordi Gumà 

Academic Editor

PLOS ONE